inorganic chemistry/chemical biology

platinum complex, 20S proteasome, inhibitor, kinetics

**Authors for correspondence:**
Tatsuto Kiwada
e-mail: kiwada@p.kanazawa-u.ac.jp
Akira Odani
e-mail: odani@p.kanazawa-u.ac.jp

This article has been edited by the Royal Society of Chemistry, including the commissioning, peer review process and editorial aspects up to the point of acceptance.

# Characterization of platinum(II) complexes exhibiting inhibitory activity against the 20S proteasome

Tatsuto Kiwada[1], Hiromu Katakasu[2], Serina Okumura[3] and Akira Odani[1]

[1]Faculty of Pharmacy, Institute of Medical, Pharmaceutical and Health Sciences, [2]School of Pharmaceutical Sciences, College of Medical, Pharmaceutical and Health Sciences, and [3]School of Pharmacy, College of Medical, Pharmaceutical and Health Sciences, Kanazawa University, Kakuma-machi, Kanazawa 920-1192, Japan

TK, 0000-0002-4928-5620; AO, 0000-0003-3786-6740

Proteasome inhibitors are useful for biochemical research and clinical treatment. In our previous study, we reported that the 4N-coordinated platinum complexes with anthracenyl ring and heterocycle exhibited proteasome-inhibitory activity. In the present study, the structure–activity relationships and characterization of these complexes were determined for the elucidation of the role of aromatic ligands. Lineweaver–Burk analysis revealed that the chemical structure of heterocycles affects the binding mode of platinum complexes. Platinum complexes with anthracenyl ring and pyridine showed competitive inhibition, although platinum complexes with anthracenyl ring and phenanthroline showed non-competitive inhibition. The structure–activity relationships demonstrated that anthracenyl moiety plays a crucial role in proteasome-inhibitory activity. The platinum complexes with naphthyl or phenyl rings exhibited lower inhibitory activities than the platinum complex with anthracenyl ring. The reactivity with N-acetylcysteine varied according to the chemical structure of complexes.

## 1. Introduction

Chemical methods for the assembly of metal complexes are based on coordinate bonds and weak interactions. Weak interactions such as electrostatic bonding, hydrogen bonding and aromatic ring stacking play vital roles in molecular recognition *in vivo*. In particular, non-covalent interactions involving aromatic rings (e.g. aromatic–aromatic interactions and hydrophobic interactions) have many characteristics and are very interesting. Aromatic rings play

crucial roles in the structure and function of proteins [1–5]. In the metalloprotein plastocyanin, the structure and the activity of the active centre are supported by non-covalent interactions of the aromatic rings [6]. Furthermore, the roles of aromatic rings have been investigated to elucidate protein–ligand interactions [7–9]. The opioid receptor ligands endomorphin 1 and 2 contain phenylalanine, and the aromatic ring is important for the affinity between endomorphin and the opioid receptor. In X-ray crystal structure analysis of protein-small molecule complexes, non-covalent interactions were observed between the protein and small molecules. Aricept, which is used in treating Alzheimer's disease, binds to a hydrophobic pocket of acetylcholinesterase [10,11]. Altogether, these findings suggested that aromatic rings play a crucial role in molecular recognition for protein–ligand interactions.

One of the proteins that exhibits an affinity for aromatic ligands is the proteasome, which plays a critical role in the regulated degradation of proteins involved in proliferation, the immune system, differentiation, cell cycle control and tumour growth [12–15]. Proteasomes possess three catalytic subunits: $\beta$1 with peptidylglutamyl-peptide hydrolysing (PGPH) activity, $\beta$2 with trypsin-like (TL) activity and $\beta$5 with chymotrypsin-like (ChTL) activity. ChTL activity is a rate-limiting step of protein degradation [16]. Bortezomib, the first proteasome inhibitor to be developed, is based on the peptide sequence recognized by the $\beta$5, which recognizes aromatic amino acids. Many inhibitors against proteasomes contain aromatic rings, and these aromatic rings bind to the pocket of proteasomes. Bortezomib and other representative inhibitors show competitive binding. However, non-competitive inhibitors have also been reported, in which the $\alpha$ subunit is used as the binding site [17–20]. Proteasome inhibitors have been investigated as anti-cancer drugs [21,22], seeds of drugs [23,24] and tools for biological study [25,26].

The platinum complex cisplatin is representative of platinum anti-cancer reagents. Its primary mechanism of action involves binding to DNA to inhibit replication. However, cisplatin also exhibits proteasome-inhibitory activities [27]. Marini and co-workers investigated the inhibitory activity of cisplatin against proteasomes and the enhancement of this activity by the proteasome inhibitor lactacystin.

Platinum intercalators, which also influence protein activities, have been investigated along with cisplatin-type platinum complexes [28–30]. Meggers and co-workers [31] reported a platinum complex that targets the protein kinase GSK-3$\alpha$. They developed a platinum complex with high inhibition activity on the basis of the non-selective protein kinase inhibitor staurosporine, in which the carbohydrate moiety was replaced with two pyridine derivatives. The synthesized complex exhibited a similar affinity to that of staurosporine, which was owing to the hydrophobic interactions and hydrogen bonding between the globular ribose binding site and the complex. These results demonstrated that a platinum inhibitor against proteins could be designed on the basis of non-covalent interactions.

In our previous report, we demonstrated that 4N-coordinated platinum complexes exhibited 20S proteasome-inhibitory activities [32]. Platinum complexes that show inhibitory activity against enzymes are rare. To the best of our knowledge, no other report has shown that 4N-coordinated platinum complexes exhibit proteasome-inhibitory activities. Our complexes are candidates for a new type of proteasome inhibitor. Therefore, in this study, we investigated the effect of ligand structure on 20S proteasome-inhibitory activity.

# 2. Results and discussion

The chemical structures of platinum complexes are shown in scheme 1. These complexes contained N-arylmethyl-1,2-ethanediamine and heterocycles as ligands. Herein, we evaluated the contribution of the ligand to proteasome-inhibitory activity. We previously synthesized complex **1** and determined its X-ray crystal structure [33]. Complexes **2**, **3**, **6** and **7** were designed as model systems of the interaction between *cis*-diamminedichloroplatinum (II) and DNA, and the physico-chemical properties were determined [34–36]. We performed screening assays of the proteasome-inhibitory activity of new and reported platinum complexes. New complexes were designed for our study. The reported platinum complexes (**2**,**3**,**6**,**7**) were designed for a purpose other than drug development. We used these reported platinum complexes as candidates for screening assays.

## 2.1. The profile of inhibitory activity against the 20S proteasome

We first determined the profile of proteasome-inhibitory activity for complexes **1–3**. As shown in table 1, complexes **1**, **2** and **3** inhibited the ChTL activity of the 20S proteasome with IC$_{50}$ values of 0.970, 0.549 and 0.916 µM, respectively. None of these three complexes exhibited inhibitory activity against the TL activity

**Scheme 1.** Chemical structures of investigated platinum(II) complexes.

**Table 1.** The profile of inhibitory activity of complexes **1–3** against the 20S proteasome. (Each data represents mean ± s.d. ($n = 3$).)

| complex | IC$_{50}$(µM) | | |
|---|---|---|---|
| | ChTL | TL | PGPH |
| **1** | $0.970 \pm 0.021$ | >10 | $0.448 \pm 0.049$ |
| **2** | $0.549 \pm 0.092$ | >10 | $0.323 \pm 0.023$ |
| **3** | $0.916 \pm 0.083$ | >10 | $0.301 \pm 0.039$ |

of the 20S proteasome. The inhibitory activity against PGPH activity was also similar among all the three complexes.

In addition, off-target inhibitory activities against other proteases were evaluated. Some current proteasome inhibitors were found to show non-specific activity against cathepsin B and α-chymotrypsin [37,38]. In particular, cathepsin B enhances the anti-cancer activity of proteasome inhibitors through the mitochondrial pathway [39,40]. However, off-target inhibitory activities against cathepsin B and α-chymotrypsin were not detected (electronic supplementary material, figure S1).

## 2.2. Lineweaver–Burk analysis

Binding mode studies have been conducted on several inhibitors [41–43]. Determination of the binding mode provides useful information that can help propose the detailed mechanism of action of inhibitors. In order to investigate the binding mode of complexes **1**, **2** and **3**, Lineweaver–Burk plots for the initial velocities of the 20S proteasome were derived under various concentrations of the substrate, Suc-LLVY of 20S proteasome and complexes **1**, **2** and **3** [44,45]. The linear plot of complex **1** intersected at $1/S = 0$, demonstrating that complex **1** binds competitively with respect to Suc-LLVY (figure 1*a*). By contrast, the linear plot of complex **2** intersected at $1/v = 0$, demonstrating that the binding mode of complex **2** is non-competitive (figure 1*b*). The binding mode of complex **3** was suggested to involve mixed inhibition (figure 1*c*). In a previous study, it was shown that heterocycles of complex **1** were non-planar and those of complexes **2** and **3** were planar. It is assumed that the chemical structure (planarity and number of aromatic rings) of heterocycles affected the binding mode of platinum complexes.

The above results suggest that complex **1** binds to the active site of the 20S proteasome. Complex **1** does not have any functional groups such as the boronic acids of bortezomib. A certain drug showed tight-binding by non-covalent interaction [31,46]. There is a possibility that complex **1** binds to the proteasome with a similar mechanism. Based on the X-ray structure analysis of the proteasome–

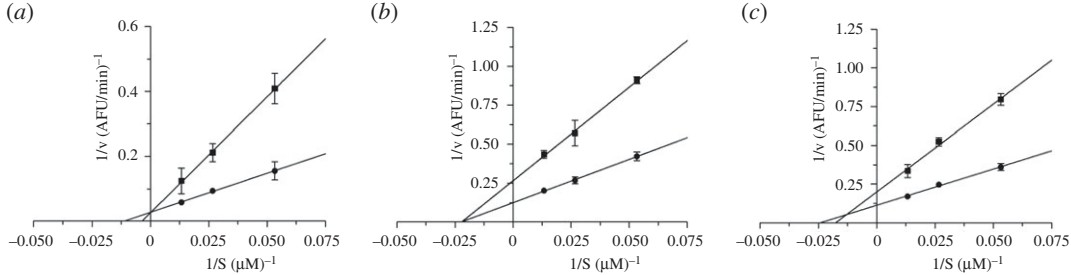

**Figure 1.** Lineweaver–Burk plot analysis of inhibitory activity of platinum complexes against the purified 20S proteasome (100 ng). (*a*) Complex **1**, (*b*) complex **2** and (*c*) complex **3** were added at concentrations of (■) 1 μM, (●) 0.5 μM. The reaction mixture was incubated at 37°C for 60 min. Reaction buffer: 25 mM HEPES–0.5 mM EDTA–0.03% SDS (pH 7.8). Each point represents mean ± s.d. ($n = 3$).

bortezomib complex, the threonine of the active site may have potentially cleaved the Pt-N bond and bound to the Pt atom of complex **1**. In a proteasome, the threonine on the active site is highly nucleophilic.

The binding modes of complexes **2** and **3** suggested that they bind to the allosteric sites [41,47]. The reported allosteric site of the 20S proteasome is an $\alpha$ ring of the core particle [17–20]. Complexes **2** and **3** may bind to the $\alpha$ ring. Complexes **2** and **3** have a positive charge. It is possible that the positive charges of complexes **2** and **3** contribute to the binding affinity with the $\alpha$ ring [20]. In addition, hydrogen bonds and hydrophobic interaction may increase the affinity [31]. By contrast, as an unusual case, inhibitors showing mixed inhibition (including non-competitive inhibition) may also bind to the active site. However, in this case, a structural change of the active site is induced (e.g. structural change associated with the catalysis of substrates) [48]. Therefore, it is thought that inhibitors recognize the difference of structure induced by the action of enzymes.

Another possible binding site of complexes **2** and **3** is the catalytic site of PGPH activity. The binding of substrates to the PGPH sites induces the inhibition of ChTL activity by allosteric regulation between the ChTL sites and the PGPH sites [49]. The binding of complexes **2** and **3** to the PGPH site may induce inhibition of ChTL activity.

It is possible that the competitive inhibition (such as complex **1**) can result from allosteric interference, which represents as an unusual case [50]. However, in that case, effects on the conformation of a proteasome are different among complexes **1**, **2** and **3**.

## 2.3. Effect of the anthracenyl moiety

The effect of the anthracenyl moiety on 20S proteasome-inhibitory activity was determined. Complexes **1**, **4** and **5**, which carried differing anthracenyl moieties, had different inhibitory activities against ChTL activity of the 20S proteasome with $IC_{50}$ values of 0.971, 2.654 and 14.21 μM respectively (figure 2*a*). These results showed that the anthracenyl moiety played a crucial role in the binding affinity to the 20S proteasome. Similarly, the anthracenyl moiety was also found to be crucial for the proteasome-inhibitory activity of complex **2**. Complexes **6** and **7** did not show inhibitory activities against the 20S proteasome (figure 2*b*). In addition, although $Pt(py)_2$(ethylenediamine) exhibited no inhibitory activity (electronic supplementary material, figure S2), N-9-anthracenylmethyl-1,2-ethanediamine itself exhibited weak inhibitory activity against the 20S proteasome (residual activity of ChTL at 10 μM was 43%; electronic supplementary material, figure S2). It is assumed that the proteasome-inhibitory activity of N-9-anthracenylmethyl-1,2-ethanediamine is enhanced by complexation with $Pt(py)_2$.

There is little difference between the inhibitory activities of complexes **1–3** against ChTL activity (table 1). These results suggest that the chemical structure of aromatic heterocycles is not important for binding affinity. It is possible that Pt atom (or positive charge of the Pt-complex) may interact with the 20S proteasome as well as the anthracenyl moiety. In such a case, the binding mechanism of these Pt-complexes may be multiple non-covalent bonding (e.g. aromatic–aromatic interaction, electrostatic interaction, hydrogen bond).

## 2.4. Inhibitory activity against the 20S proteasome in cell-free system

In molecular cell biology, the proteasome inhibitor was used in the cell lysate [51]. Inhibitory activities against the 20S proteasome in cell lysates were evaluated. Complexes **1**, **2** and **3** inhibited ChTL

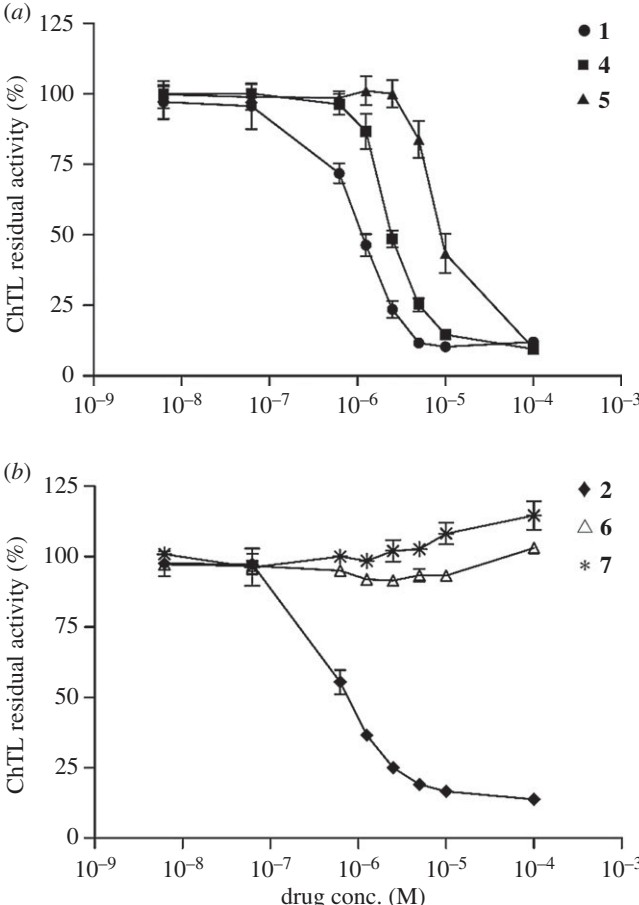

**Figure 2.** Inhibitory activities of (a) complexes **1,4,5** and (b) complexes **2,6,7** against ChTL activity of the purified human 20S proteasome (100 ng). The reaction mixture was incubated at 37°C for 60 min. Reaction buffer: 25 mM HEPES–0.5 mM EDTA–0.03% SDS (pH7.8). Each point represents mean ± s.d. ($n = 3$).

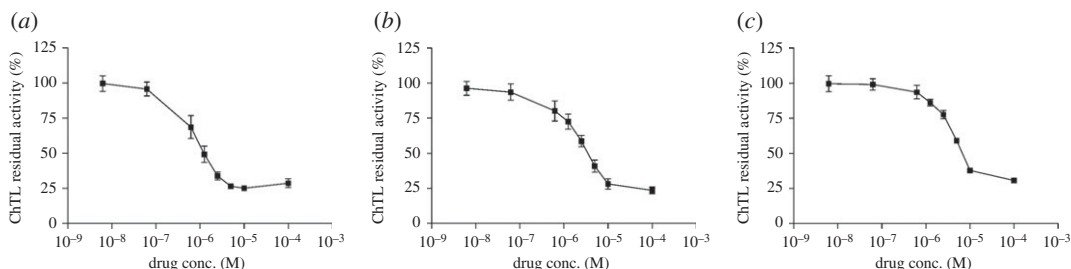

**Figure 3.** Inhibitory activities of (a) complex **1**, (b) complex **2** and (c) complex **3** against ChTL activity of the 20S proteasome from cell lysates. Each point represents mean ± s.d. ($n = 3$).

activity in the lysates with $IC_{50}$ values of 0.67 μM, 2.35 μM and 4.18 μM, respectively (figure 3). All three complexes showed proteasome inhibition at $IC_{50}$ in the micro-molar range. These results supported the hypothesis that the anthracenyl moiety is more important than the heterocyclic moiety for binding affinity. In addition, these results also showed that complexes **1–3** showed proteasome inhibition in the presence of other proteins and DNA that may show non-specific binding. As a negative control, the inhibition activity of N-9-anthracenylmethyl-1,2-ethanediamine against the proteasome was determined. N-9-anthracenylmethyl-1,2-ethanediamine showed no inhibition in the cell-free system (electronic supplementary material, figure S3).

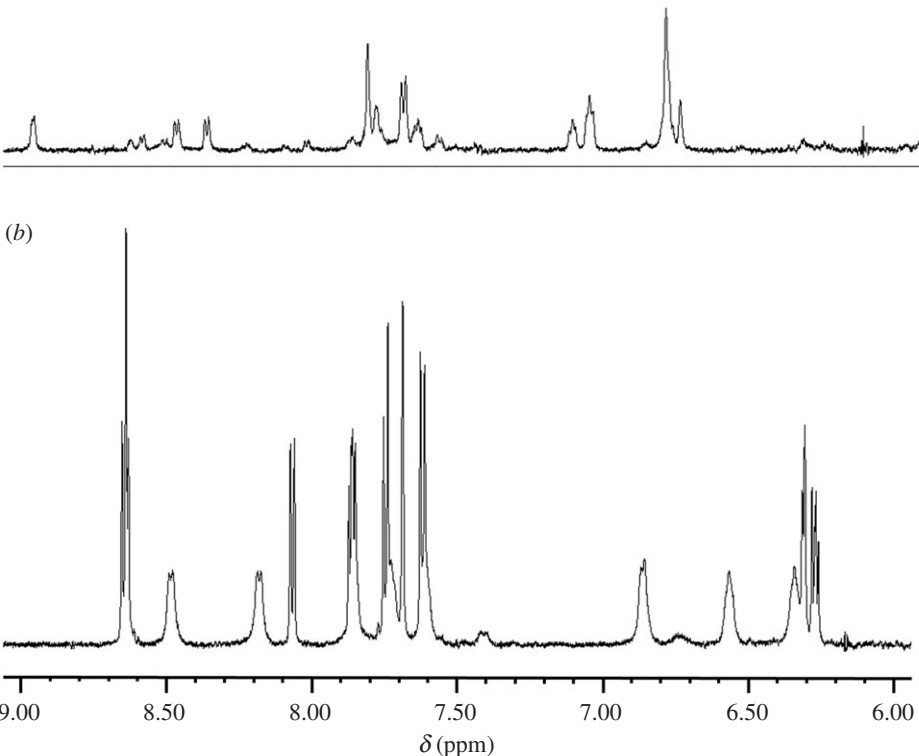

*(a)*

*(b)*

$\delta$ (ppm)

**Figure 4.** $^1$H-NMR spectra of reaction of complex **2** (10 mM) with N-acetylcysteine (10 mM). (*a*) after addition of N-acetylcysteine. (*b*) before addition of N-acetylcysteine.

## 2.5. Stability assay

Metallothionein and glutathione S transferase are representative proteins for the detoxification of heavy metals, [52,53]; thus, the reactivity of metal complexes with thiol has been investigated as a model of metal detoxification [54]. Therefore, we assessed the stability of the complexes against thiol. A Pt–N bond is principally inert. However, it was reported that cysteine could cleave Pt–N bonds. Accordingly, we determined the reactivity between N-acetylcysteine and complexes **1** and **2**. After incubation of complex **2** with N-acetylcysteine in 50 mM sodium phosphate buffer (pH 7.0) at 25°C, an orange precipitate was formed, and the nuclear magnetic resonance (NMR) spectrum showed a drastic change from the original spectrum of complex **2** (figure 4). By contrast, there was no change in the NMR spectrum of the mixture of complex **1** following incubation with N-acetylcysteine.

To identify the precipitates, the electrospray ionization (ESI) mass spectrum of the reaction mixture of complex **2** and N-acetylcysteine was measured. We could observe a signal corresponding to 1 : 1 adduct of complex **2** with N-acetylcysteine (figure 5*a*). The isotopic pattern of the signal agreed well with the calculated isotopic pattern of the 1 : 1 adduct (figure 5*b*). It is assumed that one of the Pt–N bonds was cleaved, resulting in a Pt–S bond (electronic supplementary material, figure S4).

Interestingly, complexes **1** and **2** exhibited markedly different reactivities with thiol. Similar phenomena have also been observed for other platinum complexes [54], although similar reports are very rare. These results suggested that the electronic state of the platinum coordination plane differs between complexes **1** and **2**. Complexes **1** and **2** exhibited a stacking interaction between ligands [33,36], and the stability of the stack is assumed to be correlated with the surface area of the aromatic rings. There is the possibility that stacking interaction between phenanthroline and anthracenyl moiety induced electron withdrawal from the platinum atom to phenanthroline [55]. This may induce electronic instability in the complexes. In addition, these results suggested that complex **2** is metabolized by metallothionein and glutathione. Because the amount of metallothionein and glutathione depends on the cell species, the degradation of

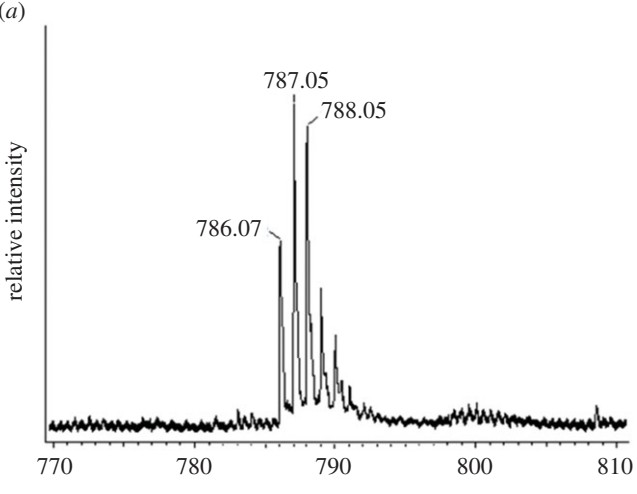

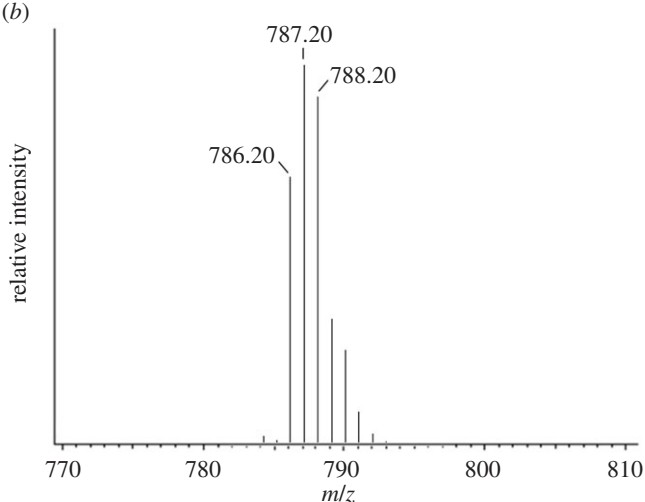

**Figure 5.** (a) Signal of ESI mass corresponding to the complex **2**-N-acetylcysteine adduct. (b) Calculated isotopic pattern of complex **2**-N-acetylcysteine adduct.

complex **2** might also depend on the cell species. On the other hand, complex **1** is assumed to be resistant to the nucleophilicity of thiol, and thiol resistance is desirable in an anti-cancer drug.

# 3. Conclusion

In this study, structure–activity relationships were examined and new and previously synthesized 4N-coordinated platinum complexes were characterized with respect to 20S proteasome inhibition. The results showed that the anthracenyl moiety played a crucial role in binding with the 20S proteasome, while the structure of the heterocyclic moiety influenced the binding mode, thereby also having a critical role in the affinity with 20S proteasome. The binding mode of complex **1** is competitive, whereas the binding mode of complexes **2** and **3** are non-competitive and mixed inhibition, respectively. Non-competitive inhibitors against the 20S proteasome have been previously reported [17–20]. The difference in the binding mode is assumed to be attributed to the chemical structure of heterocyclic ligands. The structure of the heterocyclic ligands may be recognized by the proteasome. The reactivity with the thiol group differed between complexes **1** and **2**, which probably reflected the differences in the electronic state between the two complexes. The stacking interaction between phenanthroline and the anthracenyl ring may induce withdrawal of the electron density from the platinum atom to phenanthroline. The increase in positive charge may induce polarity between the platinum atom and the ancillary ligand.

In this study, we successfully modulated the biological character of platinum complexes through changes in the chemical structure of heterocyclic ligands and ancillary ligands.

# 4. Material and methods

## 4.1. General procedures

All commercial reagents and solvents were used without purification. 2,2'-Bipyridine and 9-anthracenecarbaldehyde were purchased from Tokyo Kasei Co. Ltd. $K_2PtCl_4$ was purchased from Wako Co. Ltd. Complexes **1–3**, **6** and **7** were synthesized according to the literature with slight modification [32–35]. The human erythrocyte 20S proteasome was obtained from Boston Biochem. Human cathepsin B was obtained from BioVision, and α-chymotrypsin was obtained from Worthington Biochem Co. Ltd. Fluorescence was measured by using a MTP-880 Lab fluorescence microplate reader. $^1H$ NMR spectra were recorded on JNM ECA600/ECS400. ESI mass spectra were obtained by JMS-T100TD.

## 4.2. Synthesis

### 4.2.1. Synthesis of complex **4**

Complex **4** was synthesized according to the same method used for the synthesis of complex **1** as reported previously except that N-(1-naphthylmethyl)-1,2-ethanediamine · 2HCl was used instead of N-(9-anthracenylmethyl)-1,2-ethanediamine · 2HCl. To a suspension of $Pt(py)_2Cl_2$ (212 mg, 0.5 mmol) in $H_2O$ (20 ml), N-(1-naphthylmethyl)-1,2-ethanediamine · 2HCl (164 mg, 0.6 mmol) and $Na_2CO_3$ (74 mg, 0.7 mmol) were added. The mixture was stirred at 80°C for 16 h. The solution was filtered immediately, and the filtrate was concentrated under vacuum. EtOH was added, and white precipitate was removed by filtration. The filtrate was concentrated under vacuum and acetone was added. The pale-yellow powder was obtained. The powder was reprecipitated from water/acetone. Yield: 170 mg (50%). Elemental analysis: found, C 41.04%; H 4.76%; N 7.93%. Calcd for $[C_{23}H_{26}N_4Cl_2Pt \cdot 3H_2O]$, C 40.71%; H 4.75%; N 8.26%. $^1H$ NMR ($D_2O$), 8.55 (dd, J = 6.6, 1.8 Hz, 1H), 8.34 (dd, J = 17.1, 8.7 Hz, 1H), 8.16–8.01 (m, 1H), 7.97 (m 1H), 7.88 (m, 1H), 7.82–7.73 (m, 2H), 7.75–7.69 (br, 1H), 7.70–7.60 (m, 3H), 7.57 (m, 2H), 7.49 (m, 2H), 7.37 (m, 1H), 6.91 (t, J = 7.2 Hz, 1H), 4.69 (d, J = 13.2 Hz, 1H), 4.36 (d, J = 13.2 Hz, 1H), 3.44 (m, 2H), 3.16 (d, J = 13.8 Hz, 1H), 3.00 (d, J = 5.4 Hz, 1H). ESI-MS (m/z): 588.02 $[M-Cl]^+$

### 4.2.2. Synthesis of complex **5**

To a suspension of $Pt(py)_2Cl_2$ (212 mg, 0.5 mmol) in $H_2O$ (20 ml), N-benzyl-1,2-ethanediamine (75 mg, 0.5 mmol) was added. The mixture was stirred at 80°C for 16 h. The solution was filtered immediately, and the filtrate was concentrated under vacuum. Acetone was added, and the pale-yellow powder was obtained. The powder was reprecipitated from water/acetone. Yield: 190 mg (62%). Elemental analysis: found, C 37.75%; H 4.80%; N 8.79%. Calcd for $[C_{19}H_{24}N_4Cl_2Pt \cdot 2H_2O]$, C 37.38%; H 4.62%; N 9.18%. $^1H$ NMR ($D_2O$), δ: 8.71 (d, J = 4.8 Hz, 1H), 8.82–8.53 (br, 1H), 8.55 (d, J = 5.6 Hz, 1H), 8.07–7.88 (m 1H), 7.70–7.46 (br, 1H), 7.63–7.47 (m, 5H), 7.47–7.37 (t, J = 7 Hz, 2H), 7.37–7.25 (m, 3H), 4.01(m, 2H), 3.05(m, 1H), 2.84(m, 1H), 2.62(m, 2H). ESI-MS (m/z): 538.14 $[M-Cl]^+$

## 4.3. Purified 20S proteasome activity assay

For the proteasome inhibition assays, the peptide-7-amino-4-methylcoumarin (AMC) substrates (25 μM Suc-LLVY-AMC) and inhibitors were added to assay solutions. The assay buffer had the following composition: 25 mM HEPES, 0.5 mM EDTA, 0.03% SDS (pH 8.0). Human erythrocyte 20S proteasome (100 ng) was added to the assay buffer containing substrates and inhibitors at a final volume of 100 μl. After incubation at 37°C for 1 h, the fluorescence emission spectrum at 450 nm ($\lambda_{ex}$, 365 nm) was measured by using a fluorescence microplate reader. The inhibition against PGPH activity of the 20S proteasome was determined as described above except that Z-LLE-AMC was used as the substrate. For the assay of inhibition against the trypsin-like activity of complex **2**, the SDS was omitted from the assay buffer and Boc-LRR-AMC was used as the substrate. The assay of inhibition against the trypsin-like activity of complexes **1** and **3** was carried out by the Screening Committee of Anticancer Drugs (Cancer Chemotherapy Center, Japanese Foundation for Cancer Research).

## 4.4. Lineweaver–Burk analysis

Experimental conditions were the same as described above except that the Suc-LLVY-AMC concentration was also varied. At each concentration of platinum complexes (0.5 µM, 1.0 µM), enzyme activity was measured at various concentrations of Suc-LLVY-AMC (75 µM, 38 µM, and 19 µM).

## 4.5. Cell culture and preparation of whole-cell lysates

Cultures of colon26 were grown at 37°C in RPMI1640 supplemented with 10% fetal bovine serum. Whole-cell lysates were prepared according to the method in the literature [56].

## 4.6. Inhibition against chymotrypsin-like activity in whole-cell lysates

For the determination of inhibitory activity in whole-cell lysates, peptide-AMC substrates (50 µM Suc-LLVY-AMC), inhibitors and the whole-cell lysate (15 µg) were added to the assay solutions. The following assay buffer was used: 25 mM HEPES, 0.5 mM EDTA, 0.03% SDS (pH 8.0). After incubation at 37°C for 4 h, the fluorescence emission was measured at 450 nm ($\lambda$ex = 365 nm) by using a fluorescence microplate reader.

## 4.7. Stability assay

In 50 mM of sodium phosphate (pH 7.0), 10 mM platinum complexes were incubated with 10 mM N-acetylcysteine at 25°C. The products were analysed by $^1$H-NMR spectra and ESI mass spectra. For the measurements of mass spectra, precipitates were completely dissolved in MeOH.

Data accessibility. Data are available from the Dryad Digital Repository: https://doi.org/10.5061/dryad.s4mw6m949 [57].
Authors' contributions. A.O. conceived the project; T.K. and A.O. designed the experiments; T.K., H.K. and S.O. performed the experiments and analysed the data; T.K. and A.O. drafted the manuscript.
Competing interests. We declare we have no competing interests.
Funding. This work was partially supported by the Grant-in-Aid for Scientific Research from the Ministry of Education, Culture, Sports, Science and Technology in Japan (MEXT) (grant no. 25288027) and by the Hokkoku Cancer Funding.
Acknowledgments. Screening assays were supported by the Screening Committee of Anticancer Drugs supported by Grant-in-Aid for Scientific Research on Innovative Areas, Scientific Support Programs for Cancer Research, from MEXT.

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
