## [Reviewer comments · Royal Society Open Science]

Review History

RSOS-200545.R0 (Original submission)

Review form: Reviewer 1

Is the manuscript scientifically sound in its present form?

Yes

Are the interpretations and conclusions justified by the results?

Yes

Is the language acceptable?

Yes

Do you have any ethical concerns with this paper?

No

Have you any concerns about statistical analyses in this paper?

No

Recommendation?

Major revision is needed (please make suggestions in comments)

Comments to the Author(s)

Kiwada and co-workers reported 20S proteasome inhibitor using platinum(II) complexes. I suggest to publish this work after major revisions:

- 1) characterization of Pt complexes with protease;
- 2) give the possible binding model, binding site or docking result for this binding;
- 3) give the reasonable explanation for the good inhibitory activity;
- 4) Pls. re-write this paper. It looks like a experiment report.

Review form: Reviewer 2

Is the manuscript scientifically sound in its present form?

Yes

Are the interpretations and conclusions justified by the results?

Yes

Is the language acceptable?

No

Do you have any ethical concerns with this paper?

No

Have you any concerns about statistical analyses in this paper?

No

Recommendation?

Accept with minor revision (please list in comments)

Comments to the Author(s)

There a few rough points of writing in the manuscript - the English is a bit rough - on p. 1 "Alzheimer's Disease" is totally mangled. An editing of the English would help significantly. Compounds 1, 4 and 5 could be drawn better so the py ring does not overlap to show (almost) a carbon bonded 5 times!

Review form: Reviewer 3

Is the manuscript scientifically sound in its present form?

Yes

Are the interpretations and conclusions justified by the results?

Yes

Is the language acceptable?

Yes

Do you have any ethical concerns with this paper?

No

Have you any concerns about statistical analyses in this paper?

No

Recommendation?

Major revision is needed (please make suggestions in comments)

Comments to the Author(s)

In this paper, the authors describe the inhibitory activity against 20S proteasome (either purified or in cells) of a series of platinum(II) complexes bearing ethylenediamine ligands substituted by aryl groups (phenyl, naphthyl and anthracyl), and studied the effect of the substitution on the biological activity. Two new complexes were prepared for this study, while the other ones have previously been synthesized. The work seems well-performed but the paper is mostly observational and little explanation is given to the experimental observations.

Some issues should be resolved before possible acceptance.

Lines 14,15, page 4: An explanation should be given for the following sentence "It is assumed that the planarity and surface area of heterocycles affected the binding mode of platinum complexes".

Scheme 1: as drawn the complexes are platinum(0). I assume the compounds are dicationic, and as such the cation should be added to the structure. This is very important, as if different, the cationic parts of the molecule could play a role in the difference in activity observed.

Figures 1 and 2: Conditions should be added

Figure 3: a third plot with another platinum complex concentration would be advisable to confirm the inhibition mechanism and to calculate inhibition constants

Summary (and elsewhere in the paper): benzene, naphthalene and anthracene should be replaced by phenyl, naphthyl and anthracyl

Synthesis of the platinum complexes: compounds are only characterized by ¹H-NMR and elemental analysis. Additional analytical data such as M/S would be advisable. Coupling constant values should be given. Yields should be given in percentage, non only in mg.

Various typos should be corrected and English language should be carefully checked: For example, anthracene (35, p2), Alzheimer's diseases (58, p2), acetone (experimental)...

Decision letter (RSOS-200545.R0)

Dear Dr Kiwada:

Title: Characterization of platinum(II) complexes exhibiting inhibitory activity against 20S proteasome
Manuscript ID: RSOS-200545

The editor assigned to your manuscript has now received comments from reviewers. We would like you to revise your paper in accordance with the referee and Subject Editor suggestions which

can be found below (not including confidential reports to the Editor). Please note this decision does not guarantee eventual acceptance.

Please submit your revised paper before 04-Jun-2020. Please note that the revision deadline will expire at 00.00am on this date. If we do not hear from you within this time then it will be assumed that the paper has been withdrawn. In exceptional circumstances, extensions may be possible if agreed with the Editorial Office in advance. We do not allow multiple rounds of revision so we urge you to make every effort to fully address all of the comments at this stage. If deemed necessary by the Editors, your manuscript will be sent back to one or more of the original reviewers for assessment. If the original reviewers are not available we may invite new reviewers.

RSC Associate Editor:
Comments to the Author:
(There are no comments.)

RSC Subject Editor:
Comments to the Author:
(There are no comments.)

Reviewers' Comments to Author:

Reviewer: 1

Comments to the Author(s)

Kiwada and co-workers reported 20S proteasome inhibitor using platinum(II) complexes. I suggest to publish this work after major revisions:

- 1) characterization of Pt complexes with protease;
- 2) give the possible binding model, binding site or docking result for this binding;
- 3) give the reasonable explanation for the good inhibitory activity;
- 4) Pls. re-write this paper. It looks like a experiment report.

Reviewer: 2

Comments to the Author(s)

There a few rough points of writing in the manuscript - the English is a bit rough - on p. 1

"Alzheimer's Disease" is totally mangled. An editing of the English would help significantly.

Compounds 1, 4 and 5 could be drawn better so the py ring does not overlap to show (almost) a carbon bonded 5 times!

Reviewer: 3

Comments to the Author(s)

In this paper, the authors describe the inhibitory activity against 20S proteasome (either purified or in cells) of a series of platinum(II) complexes bearing ethylenediamine ligands substituted by aryl groups (phenyl, naphthyl and anthracyl), and studied the effect of the substitution on the biological activity. Two new complexes were prepared for this study, while the other ones have previously been synthesized. The work seems well-performed but the paper is mostly observational and little explanation is given to the experimental observations.

Some issues should be resolved before possible acceptance.

Lines 14,15, page 4: An explanation should be given for the following sentence "It is assumed that the planarity and surface area of heterocycles affected the binding mode of platinum complexes".

Scheme 1: as drawn the complexes are platinum(0). I assume the compounds are dicationic, and as such the cation should be added to the structure. This is very important, as if different, the cationic parts of the molecule could play a role in the difference in activity observed.

Figures 1 and 2: Conditions should be added

Figure 3: a third plot with another platinum complex concentration would be advisable to confirm the inhibition mechanism and to calculate inhibition constants

Summary (and elsewhere in the paper): benzene, naphthalene and anthracene should be replaced by phenyl, naphthyl and anthracyl

Synthesis of the platinum complexes: compounds are only characterized by ¹H-NMR and elemental analysis. Additional analytical data such as M/S would be advisable. Coupling constant values should be given. Yields should be given in percentage, non only in mg.

Various typos should be corrected and English language should be carefully checked: For example, anthracene (35, p2), Alzheimer's diseases (58, p2), acetone (experimental)...

Author's Response to Decision Letter for (RSOS-200545.R0)

See Appendix A.

RSOS-200545.R1 (Revision)

Review form: Reviewer 3

Is the manuscript scientifically sound in its present form?

Yes

Are the interpretations and conclusions justified by the results?

Yes

Is the language acceptable?

Yes

Do you have any ethical concerns with this paper?

No

Have you any concerns about statistical analyses in this paper?

No

Recommendation?

Accept with minor revision (please list in comments)

Comments to the Author(s)

The paper has been considerably improved, and most of the concerns answered. However, there are still many typos such as phenantholine, phenathroline, anthryl,... through the manuscript. As for the new Scheme 1, the anions have not been included and the 2+ charge looks like the double dagger symbol used for transition states. This should be corrected.

Decision letter (RSOS-200545.R1)

Dear Dr Kiwada:

Title: Characterization of platinum(II) complexes exhibiting inhibitory activity against 20S proteasome

Manuscript ID: RSOS-200545.R1

Thank you for submitting the above manuscript to Royal Society Open Science. On behalf of the Editors and the Royal Society of Chemistry, I am pleased to inform you that your manuscript will be accepted for publication in Royal Society Open Science subject to minor revision in accordance with the referee suggestions. Please find the reviewers' comments at the end of this email.

The reviewers and handling editors have recommended publication, but also suggest some minor revisions to your manuscript. Therefore, I invite you to respond to the comments and revise your manuscript.

Because the schedule for publication is very tight, it is a condition of publication that you submit the revised version of your manuscript before 19-Jul-2020. Please note that the revision deadline will expire at 00.00am on this date. If you do not think you will be able to meet this date please let me know immediately.

Kind regards,
Dr Laura Smith
Publishing Editor, Journals

Royal Society of Chemistry
Thomas Graham House

Science Park, Milton Road
Cambridge, CB4 0WF
Royal Society Open Science - Chemistry Editorial Office

RSC Associate Editor:
Comments to the Author:
(There are no comments.)

RSC Subject Editor:
Comments to the Author:
(There are no comments.)

Reviewer comments to Author:
Reviewer: 3

Comments to the Author(s)
The paper has been considerably improved, and most of the concerns answered. However, there are still many typos such as phenantholine, phenathroline, anthryl,...through the manuscript. As for the new Scheme 1, the anions have not been included and the 2+ charge looks like the double dagger symbol used for transition states. This should be corrected.

Author's Response to Decision Letter for (RSOS-200545.R1)

See Appendix B.

Decision letter (RSOS-200545.R2)

Dear Dr Kiwada:

Title: Characterization of platinum(II) complexes exhibiting inhibitory activity against 20S proteasome
Manuscript ID: RSOS-200545.R2

It is a pleasure to accept your manuscript in its current form for publication in Royal Society Open Science. The chemistry content of Royal Society Open Science is published in collaboration with the Royal Society of Chemistry.

RSC Associate Editor
Comments to the Author:
(There are no comments.)

Reviewer(s)' Comments to Author:

Appendix A

Dear Editors and Reviewers, thank you very much for taking your time to review.

Here are responses to the reviewer comments

Reviewer 1

Comment 1:

1) characterization of Pt complexes with protease;

Response 1:

We appreciate the precious suggestions provided to improve our study. We would like to answer to this comment on the assumption that “protease = proteasome”. We agree with your recommendation and believe that characterization of Pt complexes with proteasome will offer further information to understand the mechanism of action of inhibitors. Thus, we would like to perform the characterization. We trust that X-ray structure analysis of co-crystals or NMR is generally a promising method for characterization of protein-ligand complexes. However, unfortunately, we have not succeeded in the preparation of co-crystals of Pt-complex-proteasome, and NMR is not suited for analysis of structure of our platinum compounds with proteasome because intramolecular stacking is observed in our Pt-complexes. Aromatic side chains of protein may stack with ligands in our Pt-complexes in proteasomes, and the change from “ligand - ligand” interaction to “aromatic side chain - ligand” interaction may be induced. In this case, it is hard to distinguish between the effect of aromatic side chains and that of intramolecular stacking in NMR measurement. Therefore, as the first step, we performed the binding mode study in this research. Binding mode study does not show detailed mechanisms of inhibition. However, useful information can be gained.

To the best of our knowledge, our Pt-complexes are only 4N coordinated type Pt-complexes showing 20S proteasome inhibition. We believe that our Pt-complexes are new scaffolds of proteasome inhibitory complexes. The objectives of this study are to investigate the effects of structure of each moiety on biological property of 4N coordinated Pt-complexes. As the next step of our research, we would like to try characterization of Pt-complexes with proteasome.

Comment 2:

2) give the possible binding model, binding site or docking result for this binding;

Response 2:

We have discussed about the possible binding model and the binding site in the revised

manuscript (**the revised sentences 1** (shown below): line 21-45, page 3, 2nd-5th paragraph of “section 3.2 Lineweaver-Burk analysis”, and **the revised sentences 2** (shown below): line 61 page 3 - line 3, page 4, 2nd paragraph of “section 3.3 effect of anthracyl moiety”).

The revised sentences 1 (shown in “section 3.2 Lineweaver-Burk analysis” by blue highlight)

The above results suggest that complex **1** binds to the active site of the 20S proteasome. Complex **1** does not have any functional groups such as the boronic acids of bortezomib. A certain drug showed tight-binding by non-covalent interaction [31, 46]. There is a possibility that complex **1** binds to the proteasome with a similar mechanism. Based on the X-ray structure analysis of proteasome-bortezomib complex, the threonine of the active site may have potentially cleaved Pt-N bond and bound to the Pt atom of complex **1**. In a proteasome, the threonine on the active site is highly nucleophilic.

The binding modes of **2** and **3** suggested that they bind to the allosteric sites [41,47]. The reported allosteric site of the 20S proteasome is an α ring of the core particle [17-20]. Complexes **2** and **3** may bind to the α ring. Complexes **2, 3** have a positive charge. It is possible that the positive charges of complexes **2, 3** contribute to the binding affinity to the α ring [20]. In addition, hydrogen bonds and hydrophobic interaction may increase the affinity [31]. In contrast, as unusual case, inhibitors showing mixed-inhibition (including non-competitive inhibition) may also bind to the active site. However, in this case, a structural change of the active site is induced (e.g. structural change associated with catalysis of substrates) [48]. Therefore, it is thought that inhibitors recognize the difference of structure induced by the action of enzymes.

Another possible binding site of complexes **2** and **3** is the catalytic site of PGPH activity. The binding of substrates to the PGPH sites induces the inhibition of ChTL activity by allosteric regulation between the ChTL sites and the PGPH sites [49]. The binding of complexes **2** and **3** to the PGPH site may induce inhibition of ChTL activity.

It is possible that the competitive inhibition (such as complex **1**) can result from allosteric interference, which represents as an unusual case [50]. However, in that case, effects on the conformation of a proteasome are different among complexes **1, 2, and 3**.

The revised sentences 2 (shown in “section 3.3 effect of anthracyl moiety” by blue highlight)

There is little difference between the inhibitory activities of complexes **1-3** against ChTL activity (Table 1). These results suggest that the chemical structure of aromatic heterocycles is not important for binding affinity. It is possible that Pt atom (or positive charge of the Pt-complex)

may interact with the 20S proteasome as well as the anthracyl moiety. In such a case, the binding mechanism of these Pt-complexes may be multiple non-covalent bonding (e.g. aromatic-aromatic interaction, electro static interaction, hydrogen bond).

Comment 3:

3) give the reasonable explanation for the good inhibitory activity;

Response 3:

Probably, the positive charge of complexes contributes to the binding affinity. As another possibility, our complexes may bind with the proteasome by non-covalent, tight-binding mechanism (ref 31, 46). We have included the above explanation in the revised manuscript (**the revised sentences 1 and 2**).

Comment 4:

4) Pls. re-write this paper. It looks like a experiment report.

Response 4:

We have apologized for this lapse. We have included the interpretation of data and discussion as mentioned above.

Reviewer 2

Comment 1:

There a few rough points of writing in the manuscript - the English is a bit rough - on p. 1 "Alzheimer's Disease" is totally mangled. An editing of the English would help significantly. Compounds 1, 4 and 5 could be drawn better so the py ring does not overlap to show (almost) a carbon bonded 5 times!

Response 1:

We have corrected the mistake of English and scheme 1 has been re-written.

Reviewer 3

Comments

In this paper, the authors describe the inhibitory activity against 20S proteasome (either purified or in cells) of a series of platinum(II) complexes bearing ethylenediamine ligands

substituted by aryl groups (phenyl, naphthyl and anthracyl), and studied the effect of the substitution on the biological activity. Two new complexes were prepared for this study, while the other ones have previously been synthesized. The work seems well-performed but the paper is mostly observational and little explanation is given to the experimental observations.

Some issues should be resolved before possible acceptance.

Comment 1:

Lines 14,15, page 4: An explanation should be given for the following sentence “It is assumed that the planarity and surface area of heterocycles affected the binding mode of platinum complexes”.

Response 1:

We appreciate the precious advice to improve our manuscript. In a previous study, X-ray crystal structure analysis of complex **1**, **3** was performed (ref.33, 34). Two pyridine rings of complex **1** shows a non-planar arrangement and the bipyridine ligand of complex **3** is planar (free bipyridine is non-planar but Pt-bound bipyridine is planar). The phenanthroline ligand is planar whether it is free or not. The difference between complex **2** and **3** is the number of rings. We think that these different structures induce different binding modes. An explanation was provided (**the revised sentences 3** (shown below): line 16-19, page 3, “3.2 Lineweaver-Burk Analysis”, end of 1st paragraph). We just would like to show that the chemical structure of heterocycles affected the binding mode of platinum complexes. The sentence of the previous manuscript “It is assumed that the planarity and surface area of heterocycles affected the binding mode of platinum complexes” was changed to the sentence “It is assumed that the chemical structure (planarity and number of aromatic rings) of heterocycles affected the binding mode of platinum complexes” (**the revised sentences 3**).

The revised sentences 3 (shown in “section 3.2 Lineweaver-Burk Analysis” by blue highlight)

In previous study, it was showed that heterocycles of complex **1** were non-planar and those of complexes **2** and **3** were planar. It is assumed that the chemical structure (planarity and number of aromatic rings) of heterocycles affected the binding mode of platinum complexes.

Comment 2:

Scheme 1: as drawn the complexes are platinum(0). I assume the compounds are dicationic, and as such the cation should be added to the structure. This is very important, as if different, the cationic parts of the molecule could play a role in the difference in activity observed.

Response 2:

We have corrected the scheme 1.

Comment 3:

Figures 1 and 2: Conditions should be added

Response 3:

We have included the reaction condition in the figure caption.

Comment 4:

Figure 3: a third plot with another platinum complex concentration would be advisable to confirm the inhibition mechanism and to calculate inhibition constants

Response 4:

We appreciate the precious advice to improve our study. We determined the IC_{50} of complex **3** and N-9-anthracenylmethyl-1,2-ethanediamine (negative control), and we made preliminary assumption of binding mechanism (**the revised sentences 4** (shown below): line 10-11, page 4, “section 3.4 Inhibitory activity against 20S proteasome in cell free system”). In addition, we showed the raw data of the inhibitory activity of N-9-anthracenylmethyl-1,2-ethanediamine against purified 20S proteasome. This was done because we believe that it was unfavorable to have not shown the raw data of the purified protein assay and whole cell lysate assay (the raw data of the inhibitory activity of $Pt(py)_2(en)Cl_2$ is also shown in the supplementary materials because of similar reason). Moreover, IC_{50} of N-9-anthracenylmethyl-1,2-ethanediamine was not shown as opposed to the residual activity because IC_{50} in the assay that used purified 20S proteasome was approximately 10 μM , and the plot was not an ideal sigmoidal curve.

We have data to indicate that some complexes (other than Pt) lose inhibitory activity against the enzyme in the cell lysate. In this figure, we have shown that a similar phenomenon is not induced for Pt complexes. Calculation of the binding constant in the cell lysate is difficult. However, the idea behind this figure is as mentioned above.

IC_{50} of complexes **1** was changed. This is because we used a new lysate prepared for this

revision (in addition, the amount of the lysate in each well was changed from 10 μg to 15 μg). We used same new lysate (“same” means prepared in same day, similar with the meaning of “Lot”) for the whole cell lysate assay of this revision. The state of the cells may have been slightly different from the cells that were initially used due to the effect of the passage (e.g., change of expression of proteins which absorb complex 1).

The revised sentences 4 (shown in “section 3.4 Inhibitory activity against 20S proteasome in cell free system” by blue highlight)

These results supported the hypothesis that the anthryl moiety is more important than the heterocyclic moiety for binding affinity.

Comment 5:

Summary (and elsewhere in the paper): benzene, naphthalene and anthracene should be replaced by phenyl, naphthyl and anthracyl

Response 5:

We replaced benzene, naphthalene and anthracene with phenyl, naphthyl and anthracyl

Comment 6:

Synthesis of the platinum complexes: compounds are only characterized by $^1\text{H-NMR}$ and elemental analysis. Additional analytical data such as M/S would be advisable. Coupling constant values should be given. Yields should be given in percentage, non only in mg.

Response 6:

We have included MS data, percentage of yields and coupling constants to “5.2 Synthesis”. $^1\text{H-NMR}$ of these compounds are too complex. Naphthyl- and phenyl rings shows thermal motion and pyridine ring rotation. There is a complexed ring current effect among these moving heterocyclic and aromatic rings.

Comment 7:

Various typos should be corrected and English language should be carefully checked: For example, anthracene (35, p2), Alzheimer’s diseases (58, p2), acetone (experimental)...

Response 7:

The revised manuscript has undergone language correction by an edit service, and the mistakes have been corrected.

Appendix B

July 17 2020

Dear Editors:

Thank you very much for giving good advice for our manuscript. We have revised the manuscript according to the reviewer's comments. The followings are the response for comments. We really appreciate all your help and look forward to hearing from you soon.

Sincerely yours,

Tatsuto Kiwada, Ph.D.

Professor Akira Odani, Ph.D.

Reviewer's comment

The paper has been considerably improved, and most of the concerns answered. However, there are still many typos such as phenantholine, phenathroline, anthryl,...through the manuscript. As for the new Scheme 1, the anions have not been included and the 2+ charge looks like the double dagger symbol used for transition states. This should be corrected.

Response

We have corrected the typographical errors and have revised Scheme 1. In addition, in the course of correcting the typographical errors related to "anthracyl", we changed "anthracyl" to "anthracenyl". In the first revision, we changed "anthracene" to "anthracyl" according to the reviewer's comment. However, in our manuscript, the compounds name "N-(9-Anthracenylmethyl)-1,2-ethanediamine" is included. Anthracene derivatives have several synonyms. Typically, "anthracyl" is synonymous with "anthracenyl". However, we have used "N-(9-Anthracenylmethyl)-1,2-ethanediamine" in our previous papers (ref 33; "anthracenyl" is included in the title), and "N-(9-Anthracylmethyl)-1,2-ethanediamine" is not widely used. Thus, we would like to revise "anthracyl" to "anthracenyl".